# Strengthening the community governance of healthcare services in 'fragile' settings: Evidence from Burundi and South Kivu, DR Congo

**Jean-Benoit Falisse**[1]*, **Hugues Nkengurutse**[2], **Léonard Ntakarutimana**[3]

**1** School of Social and Political Science and Edinburgh Future Institute, University of Edinburgh, Edinburgh, United Kingdom, **2** Centre d'analyse et de recherche interdisciplinaire pour le développement des grands lacs (CARID-RGL), Université du Lac Tanganyika, Burundi, **3** East African Health Research Commission, Quartier Kigobe, Bujumbura, Burundi

* jb.falisse@ed.ac.uk

**Data Availability Statement:** Supporting datasets is available on OSF DOI: 10.17605/OSF.IO/RKNXS.

## Abstract

Community governance, the direct (co-)management of public services by community members, is a popular approach to improve the quality of, and access to, healthcare services–including in so-called 'fragile' states. The effectiveness of such approach is, however, debated, and scholars and practitioners have emphasised the need to properly reflect on the contextual features that may influence social accountability interventions. We study a randomised intervention during which community-elected health facility committee members were trained on their roles and rights in the co-management of primary healthcare facilities. 328 publicly-funded health facilities of Burundi and Sud Kivu in DR Congo were followed over a period of one year. In Kivu, but not in Burundi, the intervention strengthened the position of the committee vis-à-vis the health facility nurses and affected the management of the facility. HFC members mostly focused on improving the elements most accessible to them: hiring staff and engaging in basic construction and maintenance work. Using survey data and interviews, we argue that part of the discrepancy in results between the two contexts can be explained by differences in health facilities' management (whether they primarily depend on a local church or more distant authorities) as well as different local histories of relationship to public service providers. The former affects the room available for change, while the latter affects the relevance of the citizens' committee as an acceptable way to interact with healthcare providers. No effect was found on the perceived quality of and access to services, and the committees, even when strengthened, appear disconnected from the citizens. The findings are an invitation to re-think the conditions under which bottom-up accountability mechanisms such as citizens committees can be effective in 'fragile' settings.

**Funding:** Funding was provided by Directorate-General for Development and Cooperation - EuropeAid (Grant No. 2011/262-857) for the Burundi part of the research, and Cordaid and the Civil Society Fund of the World Bank for the DRC part. The funder played no role in the design and implementation of the research (inlcuding the Study design, Data collection and analysis, Decision to publish, or Preparation of the manuscript).

**Competing interests:** The authors have declared that no competing interests exist.

## 1. Introduction

Community governance, the direct (co-)management of public services by community members, is a popular approach to improve the quality of, and access to, a wide range of basic social services in low- and middle-income countries. Governments, civil society organisations, and international aid all regularly set up, organise, and strengthen citizen committees tasked with overseeing and often directly assisting the management of primary healthcare centres, schools, water pumps, and other services that are core to the life of communities. The approach is particularly popular in so-called 'fragile' settings where the State is seen as not fully capable of delivering adequate services to its citizens; community governance has been described as a vehicle for both reconstructing countries after war [1] and ensuring decent service provision in institutionally complex contexts [2]. Evidence regarding these assumptions remains limited. Faced with heterogeneous findings, which are often coming from small-scale qualitative research, the literature has turned to exploring the contextual elements that make the community governance of services effective [3–8]. Such endeavours have often remained theoretical or focused on macro-level features.

In this paper, we focus on primary healthcare delivery, a field where participatory governance practices have been implemented for decades, often with mixed results [9, 10]. We studied a programme that sought to strengthen Health Facility Committees (HFC) in Burundi and neighbouring Sud Kivu, a province of the Democratic Republic of the Congo (DRC). It was randomly implemented in 207 Health Facilities (HF) out of the 328 publicly-funded HFs of four health zones (*zones de santés*) in Sud Kivu and seven health provinces (*provinces sanitaires*) in Burundi that together cover an estimated 4.54 million people. The HFC institution, which is made of elected community members who are expected to be the co-managers of their local HF, has existed in a similar form in both countries since the 1990s but had long ceased receiving substantial support or interest from the health authorities and aid organisations by the time the programme was implemented. The intervention we looked at consisted of training sessions of HFC members and HF chief-nurses on the role of the HFC as a co-manager of the HF and the possible tools available to fulfil this role. With the same intervention being implemented in two countries, we were provided the conditions for both a rigorous assessment of the effects of community governance on healthcare delivery and an exploration of the importance of different socio-political contexts.

We assess effects on six dimensions that echo theories of change of community governance interventions [11]. These include impacts manifesting from the (1) organisation of the committee, (2) the accountability of the health facility, (3) the management of the HF, (4) the perceived quality of, and (5) access to services, and eventually the (6) delivery of services (uptake). In Burundi, only the organisation of the HFC was significantly affected by the intervention. In Sud Kivu, the intervention led to a significant change (0.799–0.823 SD) in the HF-level accountability index and a less important but still statistically significant change (0.153–0.276 SD) in the HF management index. These effects, however, do not translate into visible changes in the indexes reflecting the provision and quality of services.

The present article follows two articles on the Burundi case [12, 13]. The interventions they focussed on–attempting to strengthen the relationship between HFC and local elite and training the HFC on acquiring and analysing HF data–came on top of the intervention described in the present article. They were, overall, not more effective; HFCs in Burundi appeared mostly stuck. In this earlier work, we had hypothesised that contextual factors such as the nature of politico-economic power relations at the local level are crucial for understanding when external programmes indeed lead to reinforcing HFCs. The stark difference in results between Burundi and South Kivu presented in this paper helps further establish which elements of the

contexts matter most. Indeed, the difference seems mostly explained by (1) the higher proportion of semiautonomous faith-based, and especially Protestant, HFs, in South Kivu; and (2) different relationships to authorities and service providers, with people displaying a more confrontational attitude in South Kivu. These findings have important implications for understanding when social accountability works [14] and when local, externally driven, pro-democratic reforms may be taken up and adopted. They also have crucial implications for the multi-billion dollar industry of social accountability interventions in healthcare and other fields.

Our main contributions are to three debates:

First, empirical, theoretical, and grey literature alike have pointed out that participatory institutions, first and foremost, require citizens to know about the functioning of the services they are expected to co-manage and to master the information and management tools at their disposal. The mainstream developmental discourse insists on the need for capacity-building. Such 'institutional knowledge' gap hypothesis [15] has, however, rarely been subjected to proper impact evaluation and has been criticized for its naivety [16]. We suggest that, under certain circumstances, focussing on knowledge and skills may be effective.

Second, the sample and area of the study allow us to look at the political economy of service delivery at the micro-level. Numerous authors [4, 5, 8] discuss 'power' and 'state-society relations', often from the perspective of the attitude of the State towards the idea of social accountability. We look at the attitude of citizens towards the State and the local spaces for collective action.

Third, by looking at the same intervention in two countries, we contribute to the broader debate on the external validity and policy relevance of randomised control trials. The problem is the narrowness and scope of evaluations and, ultimately, the portability of findings: will the mechanisms that produced effects in one place produce the same effects somewhere else [17]? By exploiting variance in the local situations within each country, we break down the context.

The next section introduces the research questions and reviews some of the most relevant literature on HFC strengthening and community governance. The third section provides more background information on the HFC and shows their limited effectiveness prior to the intervention. The fourth section presents the intervention and our mixed-methods research design. Section five analyses the main effects of the intervention and the possible reasons for the different effects found between Burundi and South Kivu.

## 2. Rationale and literature

A dense literature has accompanied the evolution of the concept of community participation in basic social services. Development studies scholars and anthropologists see community participation as a catalyst for further social change and as having value in itself because it echoes democratic principles [18]. Alongside these disciplines, development economists have gradually taken an interest in the idea of community involvement as a potentially efficient and fair way to allocate of public goods and services. It follows a straightforward utilitarian perspective: if people participate in and own their services, these services should be more in line with their needs [19]. The term 'community participation' has been gradually replaced and broadened by the ideas of 'community governance' and 'social accountability', which are now at the core of mainstream approaches in international development [20].

In healthcare, the idea of community governance emerged globally well before the 1990s push for good governance. Pilot projects, whereby communities identified problems and then proposed and implemented solutions, started in the 1960s in socialist and non-aligned countries. They led to an embedding of community participation into the 1978 Alma-Ata WHO

Declaration on primary healthcare, 'in the spirit of self-reliance and self-determination' [21]. The peculiarities of the healthcare market, namely asymmetric information, third-party agents, barriers to entry and substantial externalities [22], contributed to the legitimisation of community-based interventions. In Africa, the 1987 WHO-UNICEF Bamako Initiative further stressed the idea of community involvement in health and encouraged social mobilisation. Across the continent, citizens' committees were set up and tasked with representing the interests of the population in primary health care facilities.

A prerequisite for community participation via citizens' committees to work is that their members understand what their role is, the nature of its boundaries [14], and how the institution they are supposed to manage functions. The idea of 'institutional knowledge' [23] entails the understanding of the official regulatory framework of the committees and health facilities as well as the general rules of the game that are expected to be similar across health facilities and committees. In addition to knowing their roles, it is also expected that the committee members have the capacity to play them [15, 24]. Capacity or capability refers not only to understanding but also to 'identifying and solving self-identified problems' [25], which is, by definition, context-specific and is the core of most interventions which seek to initiate or strengthen community governance.

In spite of dense literature on community governance, there is a lack of substantial empirical evidence on the impact and efficiency of health facility committees [9, 14, 26]. Most existing studies have weak identification mechanisms and sometimes only rely on qualitative or cross-sectional data [10]. In Zimbabwe, a randomised intervention in 149 health facilities trained health centre committees on the use of community scorecards, suggestion boxes, and feedback forms [27]. It had quite a limited impact in terms of social accountability and no effects in terms of quality and use of services. The authors suggest that the committee members did not manage to ensure that community voices were truly represented in decision-making processes as they did not engage with the community. Earlier, Björkman and Svensson's randomised control trial in Uganda [28] considered HFC-facilitated meetings between the population and HF staff, which were organised to discuss the performance of the HF and define actions in health facilities. After one round of meetings, it proved to be one of the most successful community-based interventions reported (McCoy, Hall & Ridge 2011), with an improvement in indicators such as the weight of infants, under-fives mortality, and services. However, it is not possible to clearly identify what in the intervention package was so effective, and in particular, whether the effect is primarily driven by health information given to the population at the meetings rather than citizens' voice. In fact, when Raffler and colleagues [29] sought to replicate some of the intervention later in Uganda, not only did they find more modest impact, but they also they found no evidence that the citizen monitoring component of the intervention was driving the effects.

Last but not least, the literature in public health and development studies suggests that 'context matters' for the HFC to gain influence at the HF [26, 30], but often falls short of providing empirical evidence on what exactly makes the difference. This is an important area for the research agenda on community governance [31]. Looking at the broader picture of community-driven development initiatives promoted by development aid, beyond the health sector, the scant literature focusing on Burundi and the Kivus has generally concluded with mitigated results [32, 33].

## 3. Health facility committees in Burundi and South Kivu

Burundi and the DRC are ranked at the bottom of most international health and socio-economic development indexes. Their health systems are under pressure and the *centres de santé*

(HFs), the frontline primary healthcare centres, are the main providers of care to the population. Although almost always only staffed with nurses, such HFs provide services commonly associated with the intervention of a medical doctor in other countries, for instance: onsite childbirth, the treatment of tuberculosis and acute malaria, and sometimes even distributing medication for HIV/AIDS.

The HFs of Burundi and South Kivu have a significant degree of management autonomy, which has been formalised through devolution processes and accentuated by two factors in the past decades. Firstly, periods of war repeatedly deprived HFs from critical material and human resources, forcing them to cope with locally-sourced resources for years and sometimes decades. Secondly, post-war New Public Management-inspired health reforms such as performance-based financing have insisted on the autonomous management of health facilities in both settings [34].

Official guidelines and regulations at the time of our research described broadly similar structures in both countries [35, 36] (these guidelines have been updated since our study, but the general approach remains similar [37, 38]).

A chief nurse (*infirmier titulaire*) heads the HF, they are seconded by a deputy chief nurse (*titulaire adjoint*). Key inputs, including drugs and qualified nurses, are dispatched from the Ministry of Health (national level in Burundi, provincial level in DRC) via health districts, but support staff and small equipment are hired and purchased locally, respectively. The Ministries of Health cap the prices of drugs and services. Faith-based clinics (*centres de santé conventionnés* in DRC or *confessionnels* in Burundi) co-exist alongside purely public and entirely private HFs. Even though they enjoy more management autonomy than strictly public HFs in relation to staff management as well as drugs and equipment purchasing, faith-based HFs remain regulated and subsidised by the government.

Every publicly-funded HF must have an HFC. In both countries, the HFC is a similar institution that is responsible for co-managing the HF with the chief-nurse at different levels: technical (HF planning according to the health situation of the population); (2) administrative (prices monitoring, liaison with administration, inventories, etc.); (3) financial and strategic (reviews and action plans); and (4) human resources. Finally, the HFC is also tasked with contributing to the promotion of HF activities among the population. HFC members are unpaid volunteers, even though some receive a minor financial compensation from their HF. The population of their catchment area elects them. Their mandate is officially for two years in Burundi (but often lasts longer [13]). There is no official time limit in the DRC.

By the time of our study, there were strong indications, but no systematic study, suggesting that the HFC system was not effective, with HFCs only loosely involvement in the co-management of their health facilities [39, 40].

## 4. Methods

The situations of the HFCs in Burundi and South Kivu allowed for exploring the skills and knowledge gap hypothesis. It presented an opportunity to see what difference knowledgeable HFCs, equipped with all the tools and information required, make to social accountability, HF management, and service delivery. The comparative nature of the study was also a chance to try and understand how different settings may produce different outcomes.

### Ethics statement

The research was approved by Oxford University's Social Science Division (SSD/CUREC1/12-006) as well as Burundi's National Ethics Committee (*Comité National d'Ethique*, CNE, June 2012). The implementing NGO secured authorisation from the Ministry of Health (*Direction*

*Provinciale de la Santé*, DPS) in South Kivu with whom the research protocol was shared, discussed, and validated. Formal verbal consent was obtained for both the surveys and interviews (written consent was deemed inappropriate given the context marked by, among others, high levels of illiteracy).

## Intervention

After preliminary research and consultations with representatives of HFs, HFCs, and the population, an intervention was designed in collaboration with the Ministry of Health of Burundi and the Dutch NGO Cordaid, with the objective of strengthening the HFC roles and capacities. Preliminary research on HFCs in South Kivu highlighted issues similar to Burundi's, and the Ministry of Health of South Kivu decided to implement an intervention similar to Burundi's in all aspects. Burundian facilitators trained their Congolese counterparts.

The intervention took place in seven health provinces of Burundi (Bubanza, Bujumbura Rural, Bururi, Makamba, Rutana, Ruyigi, Cankuzo) and four health zones of South Kivu (Walungu, Katana, Miti-Murhesa, and Idjwi). It started with a two-day training with modules on (1) the local health system, the services offered at the HF, and its administrative functioning; (2) the roles of the HFC according to Ministry of Health and World Health Organization guidelines; and (3) the management tools used at the HF such as budgets and development plans. It also presented the HFC with tools that have been used by citizen committees, such as scorecards to prioritise actions and terms of reference to clarify the HFC internal functioning and the HFC-HF relationship. In line with the 1990s debate on community participation in health [21], the intervention provided tools and information, but community actors were left free to use them as they wanted, depending on their analysis of their own context. One booklet containing a Kirundi or Swahili translation of the legislation on the HFC was distributed in each HF, with clear instructions given to keep it at the HF. Simple language, role-plays, participant-to-participant teaching, as well as images were used to improve the chances that skills and knowledge were efficiently learned. The initial training was followed by: (1) three months later, phone contact with the HFC president to discuss the situation of the HFC; (2) six months later, a half-day recap session; and (3) nine months later, another half-day recap session. The recap sessions were discussions on the functioning and activities of the HFC.

Three aspects that are known to influence the efficiency of training sessions received particular attention: (1) material expectations were kept as low as possible by only compensating for travel expenses and a soft drink during the initial training and recap sessions; (2) potential facilitator effects were mitigated by recruiting facilitators with similar higher education and ethnic backgrounds (Hutu and Tutsi in Burundi, Mashi and Muhavu in South Kivu) as well as by rotating them between initial and recap sessions; (3) the sessions were organised away from the potential interference of civil servants external to the HF, such as health district officials.

The present paper focuses on the pilot phase of the implementation of the intervention, which was randomly implemented in in 168 pilot HFs/HFCs of Burundi and 39 pilot HFs/HFCs of South Kivu (using the random draw that minimized differences between the control and pilot group out of 10,000 random draws). 122 HFs/HFCs (83 in Burundi and 39 in South Kivu) were not exposed to the intervention during the pilot phase and constitute a control group. In Burundi, the split between control and pilot is not 50/50 because the pilot group included different subgroups (three interventions building on top of the basic intervention were implemented in Burundi and are not reported in this paper [see 12, 13]. They do not affect the findings of the present paper, they are orthogonal to the intervention and fully taken into account in the estimates). The Randomised Control Trial (RCT) started in July-October 2012 in Burundi and in September-October 2012 in South Kivu and was evaluated after a year.

## Data sources and outcomes

Measuring changes in community governance and delivery of services is not straightforward. The number of potential outcome variables is important, especially when bearing in mind the limited predictability of institutional reforms [41]. Our approach follows other work in the field of reform in local institutions [42]. We derive a set of outcome variables from the HFC theory of change presented in official documents (MoH, Burundi & WHO 2007) according to which HFC training is expected to modify: (1) the HFC organisation and (2) HF accountability, which leads to changes in (3) management, in turn improving the (4) access, (5) perceived quality, and (6) use of healthcare service. Because alternative causal pathways are always possible, we look at these six aspects one by one. Table 1 summarises the key indicators used to study the different dimensions and S1 Table provides more details on the data sources for each dimension and the sample size.

The list was established pre-intervention (the plan was presented as part of the PhD work of one of the authors but not formally registered). The first dimension seeks to capture indicators assessing whether the HFC knows what it is supposed to do and is organised in accordance to official guidelines. The second dimension is social accountability and is operationalised following the key components defined by Brinkerhoff [43]: enforcement and sanction mechanisms (who decides on what), information given to users, and circulation of information through the HFC. The third dimension, HF management, is constructed based on the theoretical production function of an HF: (1) human capital (medical staff), (2) drugs and equipment, (3) infrastructure, and (4) finances, including which services are available. Finally, the delivery of services was examined through data on the perceived quality of services, household-level measures of access to healthcare, and HF-level activity indicators. Our focus is on dimensions rather than individual indicators. A mean standardised score was calculated for each dimension introduced in the previous sub-section, following the approach described by Glennerster & Takavarasha [44].

The data originates from different sources. Routine health information system data was available for all HF and provided monthly information on the use of services and yearly data on the infrastructure and human resources of each HF. It is mostly used for our dimension 6: "provision of services". Dedicated surveys with the HFC president, HF chief-nurse, and households living in the catchment area of each HF were necessary to better understand the

**Table 1. Composition of outcome indexes (details in S1 Table).**

| Dimension | Type of indicators |
|---|---|
| 1. HFC organisation | *HFC-level*: knowledge of role, organisation and decision process |
| 2. Accountability | |
| 2.1. HFC rights | *HFC-level*: decisions of HFC, HFC president, and HFC executives |
| 2.2. Share info w. users | *household-level*: information to patient, tariffs displayed, bills clarity<br>*household-level*: knows HFC, attended meetings, met HFC members |
| 3. Management | |
| 3.1. Drugs and equipment | *HF-level*: days of stock-out of main drugs and equipment |
| 3.2. Human resources | *HF-level*: staff at HF (different levels of qualification) |
| 3.3. Infrastructure | *HF-level*: building (new, repaired), electricity, water |
| 3.4. Finances | *HF-level*: services open, balance in last 3 months |
| 4. Equity in access | *household-level*: barriers to access care, denial of care |
| 5. Quality of services | *household-level*: welcoming at HF, attention given to patient |
| 6. Provision of services | *HF-level*: number of visits in core area/catchment area population |

dynamics of community participation and HFC involvement in the HF. They are used to assess our dimensions 1, 2.1, 3 (HFC organisation, HFC rights, and HF management). The data was collected by a dedicated team of enumerators who surveyed all HFC presidents and HF chief-nurses prior to the intervention and then again between October 2013 and January 2014. In addition, a random sample of 30 individuals were interviewed in each catchment area at baseline and end line in both countries, with the exception of the baseline survey in Burundi that only took place in a random selection of half the HFs of both the intervention and control groups of HFs. The household survey data is used to assess dimensions 2b (information sharing with users), as well as 4 (perceived quality of care) and 5 (equity in access to services). As S2 Table indicates, key indicators are similar between intervention and control groups. In addition to the surveys and secondary data, we organised exploratory qualitative research in 46 sites in Burundi in 2011 and eight sites in South Kivu in 2012. Semi-structured interviews and focus groups were also organised with nurses, HFC members, HFC presidents and local leaders in 60 sites in Burundi and 17 of South Kivu at the same time as the end-line survey. The qualitative data was used to refine hypotheses about the mechanisms that underpin the findings and was crucial to understand the difference in results between Burundi and South Kivu.

### Evaluation framework

The main effects of the intervention are evaluated using a standard difference-in-difference approach and standard errors are adjusted for multiple hypotheses testing (family-wise error rate). More technical information on the evaluation framework is provided in S1 Text. The main results are presented in graphical form, when possible. Details on the value of coefficients and exact significance levels are found in the supplementary tables, which also provides a range of robustness tests, including adding a set of additional control variables and using a different model specification, an analysis of covariance ANCOVA [45], when relevant.

## 5. Results and discussion

This section is divided into four sub-sections that present what is effectively a three-stage argument: subsection (a) presents the situation at baseline using our novel survey data; subsections (b) and (c) respectively present and discuss the main findings; and, finally, subsection (d) presents additional qualitative and quantitative evidence that help cast light on the main finding of the paper, which is the difference in outcomes between Burundi and South Kivu.

### a. Situation at baseline

Before we focus on the results of the evaluation, it is useful to note that our research provided further details comforting the idea that, in practice, the HFCs are often poorly functional (and thereby strengthening the rationale for the intervention). As shown in Table 2, presenting the situation at baseline, HFCs rarely named HF management among their main activities, took part in very few decisions at their HFs, and often simply had little idea of what they were supposed to do. The differences between the HFCs of South Kivu and those of Burundi are minor in economic terms. In practice, the chief nurse appeared to be the real and only chief at the HF. As one former Ministry of Health official in Burundi put it in an interview, the HF often remains 'something that belongs to the chief nurse'. What HFCs do consistently are preventative healthcare activities.

The lack of skills and knowledge of the HFC members ─on checks and balances, budget, planning, and HF functioning in general─ came as the first and main explanation in almost all the exploratory interviews with HFC members, HFC presidents, nurses, and the population.

**Table 2. HFC baseline situation.**

|  | Burundi | | South Kivu | |
|---|---|---|---|---|
|  | **mean** | **SD** | **mean** | **SD** |
| *Involvement in decision over seven key areas at the HF* (0: none; 1: advises; 2: decides)[1] |  |  |  |  |
| HFC decision rights according to HFC | 0.154 | (0.248) | 0.262 | (0.352) |
| ditto according to chief nurse | 0.178 | (0.270) | 0.200 | (0.379) |
| HFC president rights according to HFC | 0.450 | (0.356) | 0.394 | (0.478) |
| ditto according to chief nurse | 0.470 | (0.376) | 0.452 | (0.488) |
| *Top 3 activities of HFC, by category, according to HFC* |  |  |  |  |
| co-management[2]: cited first | 0.134 | (0.341) | 0.1 | (0.303) |
| second | 0.24 | (0.428) | 0 |  |
| third | 0.305 | (0.463) | 0.037 | (0.192) |
| community health and health promotion: cited first | 0.709 | (0.455) | 0.760 | (0.431) |
| second | 0.671 | (0.471) | 0.897 | (0.307) |
| third | 0.638 | (0.483) | 0.926 | (0.267) |
| n | 251 |  | 78 |  |

1. The score is the average of decision about: hire/dismiss staff, hire/dismiss support staff, order drugs, order equipment, drugs pricing, services pricing, and development plan. | 2. considered broadly, for instance, the few cases where the HFC helped the HF recover debts from patients are included here | Difference South Kivu—Burundi: statistical paired T-tests significant at p<0.1 for all variables except HFC decision rights, which is significant at p<0.05. | source: HF/HFC baseline survey presented in the methods section.

## b. Main effects of the intervention: Results

The intervention had robust and statistically significant effects on two indexes: the organisation of the HFC and the HF-level social accountability (Fig 1 and S3 Table). The effect on the HF management index is borderline significant (it is significant only in the case of the ANCOVA specification, see S3 Table). The intervention did not lead to substantial changes in the use of services, or in their perceived quality and access.

Fig 2 shows the main effects of the intervention by country using the diff-in-diff model. While the training intervention led to visible changes in the organisation of the HFC in Burundi, it led to significant changes in the HFC organisation index but also to a 0.823 SD change in HF-level accountability indicators and a 0.276 SD change in the HF management indicators in South Kivu. These findings are robust to alternative specifications ─with the noticeable exception of the effect on HFC organisation in Burundi (see S4 Table).

Looking at the individual components of the main indexes impacted by the interventions allows a finer understanding of the dynamics at play. The social accountability indicators (S5 Table) that are robustly affected by the intervention in South Kivu mostly relate to the decisions rights of the HFC at the HF (columns 1–6) and only to a more limited extent to information sharing (columns 7–10). The decision rights of the HFC ordinary members, HFC executives and HFC presidents are found to have improved when measured both from the perspective of the HFC (columns 1, 3, and 5) and from the perspective of the chief nurse (columns 2, 4, and 6). Interviews carried out in the HFCs confirm a change in the HFC-HF relationships in South Kivu mostly. In many Burundian HFCs, HFC members typically explain that they clash with HF staff over their rights, and lose.

Finally, the changes in HF management observed in South Kivu appear, as shown in Fig 3 (details in S6 Table), primarily driven by changes in (1) resources mobilisation and especially the building of new infrastructure and (2) human resources, the hiring and replacement of nurses and chief nurses.

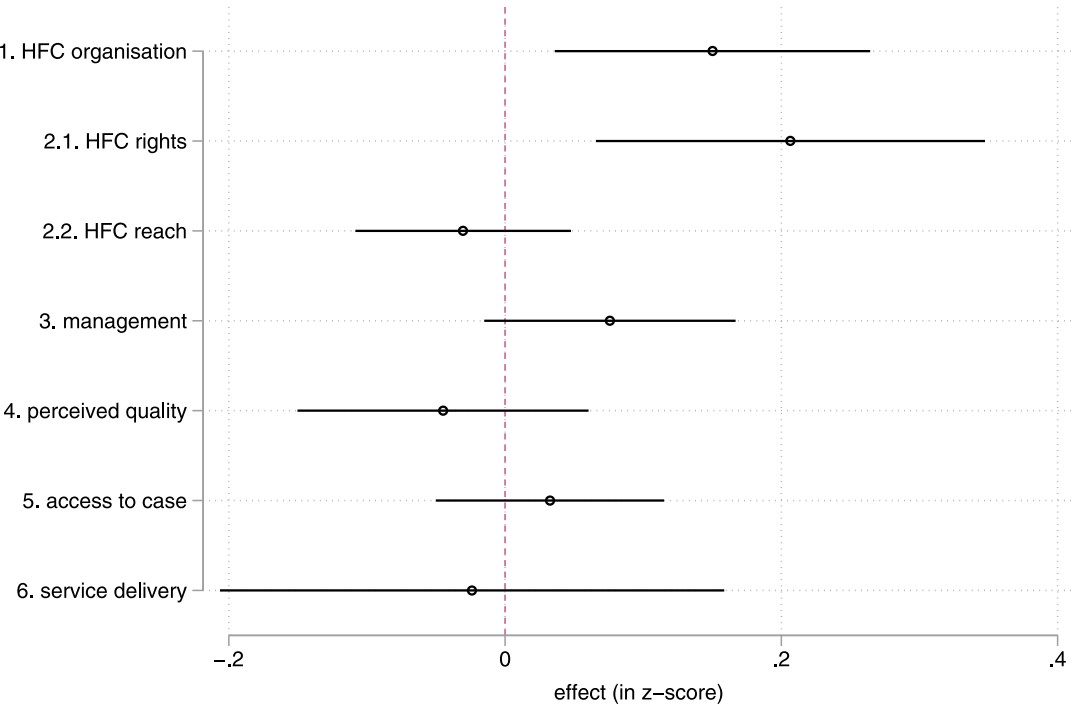

**Fig 1. Intent-to-Treat—Main effects (diff-in-diff, without controls).** Note: lines represent a 95% confidence interval for the point estimate. Estimates whose lines cross the red vertical line are not statistically different from zero. Detailed results are found in S3 Table. The standard errors presented on the chart are without FWER correction, but both positive effects are robust to such correction. | Please see the methods section and S1 Table for details on the sampling (estimates use the full sample, n = 329 HFC/HF) and data sources.

## c. Main effects of the intervention: Discussion

There was no reported issue with the implementation of the intervention: training and recap sessions all took place as expected and were well-attended. Facilitators did not have any contact with the HFCs of the control group, making spill-over effects unlikely. Moreover, the HFCs of the control group did not see any significant change in the adoption of terms of reference, one of the most direct outcomes of the intervention. The latter indicator is also, across specifications (S5 Table), driving the effect on the HFC organisation index in the case of Burundi, which suggests that what is at play in Burundi maybe be more akin to 'institutional mimicry' [46], HFC adopting the appearance expected from them, than a genuine change in functioning.

The effects on management are real in South Kivu, but also limited. The indicators that relate to more technical skills, such as the financial supervision of the HF, do not change. A possible explanation is that the HFC members deliberately invested their energy into low-hanging fruit types of activities of limited complexity on which they could have easy leverage: building new infrastructure, exerting pressure to change and hire staff, and checking for shortages of drugs.

Importantly, these changes are not associated with changes in access to, quality, and use of services. Some coefficients even take a negative, albeit not statistically significant, value (columns 4 and 6 in S3 and S4 Tables). The short time frame makes an ideal suspect for the lack of effects on service delivery and quality. Changes in management, which are not even robust in this case, may take time to have visible effects on the use of services. Moreover,

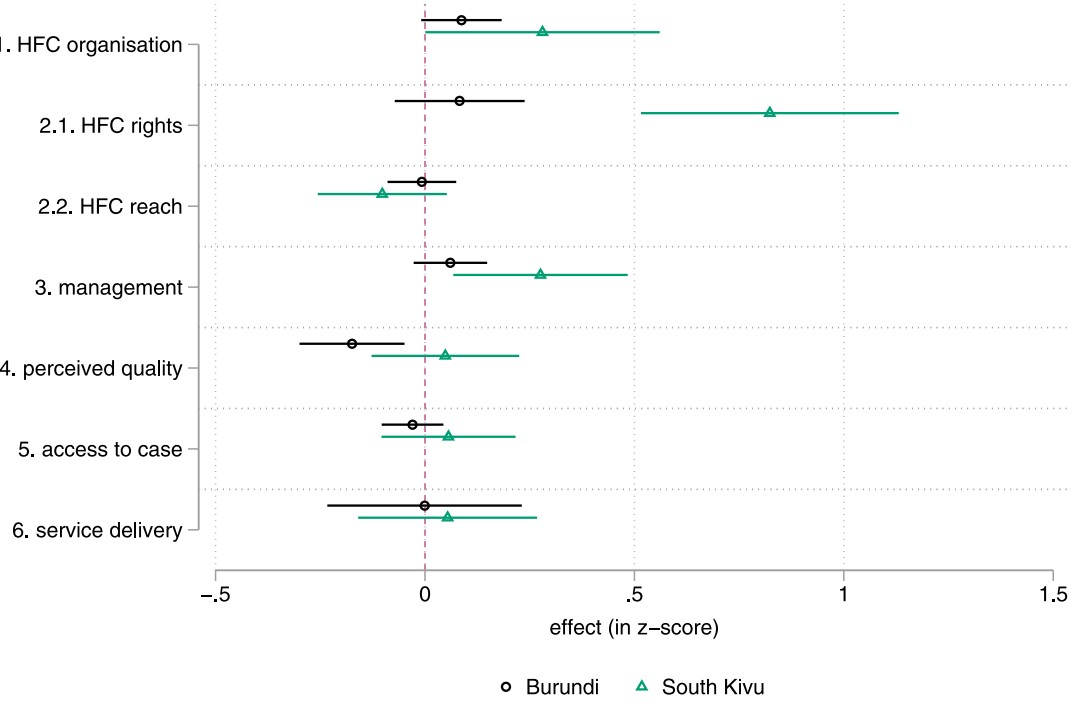

**Fig 2. Intent-to-Treat—Main effects (diff-in-diff, without controls), by country.** Note: lines represent a 95% confidence interval for the estimate. Estimates whose lines cross the red vertical line are not statistically different from zero. Detailed results are found in S4 Table. The standard errors presented on the chart are without FWER correction. Only the South Kivu effects are robust to such correction. | Please see the methods section and S2 Table for details on the sampling (estimates use the full sample, n = 329 HFC/HF) and data sources.

substantial literature on both social accountability measures and community participation in health warns against presenting community participation as a magic bullet [47]. It is, however, worth noting that our time frame is not substantially shorter than other studies that found a positive impact of community participation initiative on the use of health services in other contexts [28]. No change was observed in terms of use of services (column 6) when extending the period by nine more months (until when new interventions started being implemented in control HFCs).

More fundamentally, even the more encouraging results in South Kivu seem to point to a form of disconnect between the committee and the population, which echoes a recent impact evaluation of health committees in Zimbabwe [27]. This may be another reason for the lack of changes in the quality and use of services: there is no clear evidence that the changes at the HF are closely aligned with the most pressing needs of the population. In other terms, characterising the HFC-HF as a classic principal-agent problem, the intervention seems to improve the situation of the HFC as the principal of the HF (in South Kivu) but not necessarily of the population as the principal of the HFC. The training intervention did not provide explicit incentives for the HFC to reach out to the population, and both the HF/HFC-level and household-level indicators on this are generally not affected by the intervention (column 2.2 in S4 Table, as well as column 10 in S5 Table). By the time of the end-line survey, 26.4% of the interviewed household in South Kivu still had never heard of the HFC, and only 28.3% identified the HFC. Among them, 32.3% had not interacted at all with an HFC member in the last year (neither indicator was impacted by the intervention).

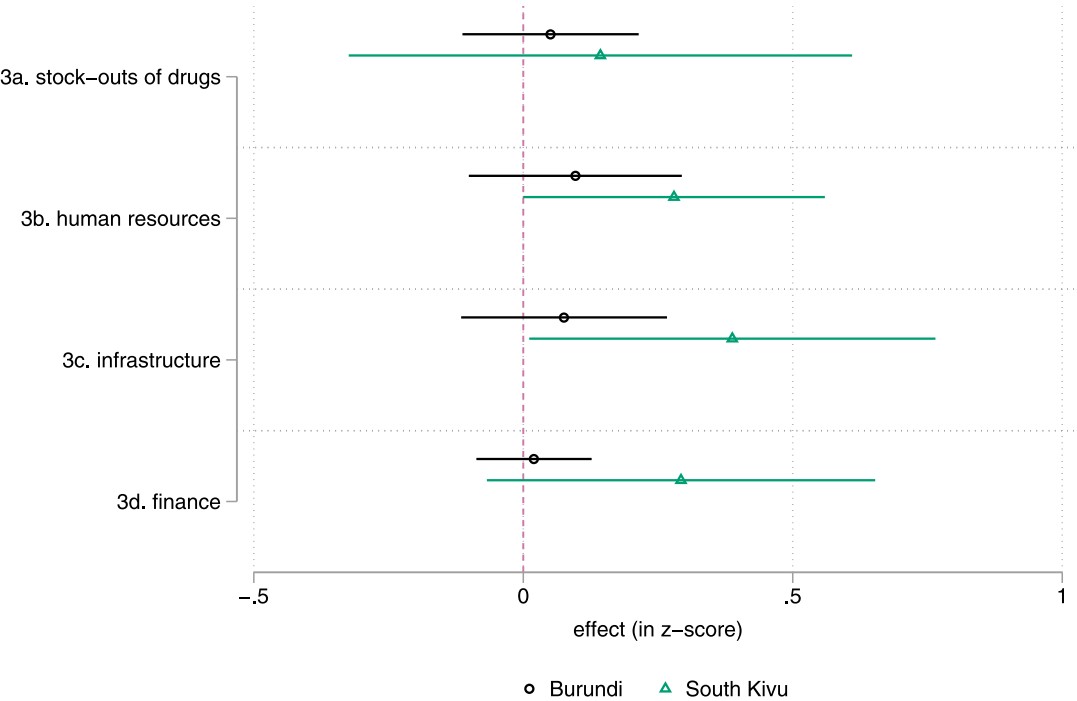

**Fig 3. Intent-to-Treat—HF management (diff-in-diff, without controls), by country.** Note: lines represent a 95% confidence interval for the estimate. Estimates whose lines cross the red vertical line are not statistically different from zero. Detailed results are found in S6 Table. | Please see the methods section and S1 Table for details on the sampling (estimates use the full sample, n = 329 HFC/HF) and data sources.

There is, however, a strong difference between South Kivu and Burundi: a simple training intervention on rights and tools did make the HFC a more influential player at the HF and even affected management indicators. This brings nuance to some of the recent literature on social accountability, which stresses that information and basic training are not enough to provoke changes [6]. In the context of South Kivu, basic training was associated with a change in HF management and the overall balance of power at the HF. Yet, the qualitative data suggests that the story behind the changes that took place in South Kivu may be driven by more than the acquisition of new critical knowledge and skills. Firstly, the intervention was also a reminder to the HFC members but also to the HF staff, of the rights and duties of the HFC. It assigned the HFC a clear role that they did not have before, or at least not explicitly. Secondly, the intervention was carried out by an external and respected NGO and legitimised the position of the HFC at the HF. The change could be in terms of the sense of entitlement (or self-perception) of the HFC (see S5 Table), or in terms of the perception other actors have of the HFC.

### d. Additional results: Beyond 'context matters' (heterogenous effects)

The stark contrast between the effects of the intervention in Burundi and in South Kivu is puzzling. Rather than testing every possible explanation, we carried out interviews (15 in total) with key informants such as health system professionals and civil society leaders on both sides of the border. Three families of hypotheses emerged from these discussions: (1) HFC membership matters: South Kivu HFC members are better educated and may, therefore, be more capable of playing their role; (2) incentive and management structures are different across the

border: Congolese HFs have more autonomy in management because a higher proportion of them are managed by local churches that give more room to the HFC; (3) local relationships to public service providers and the State are different. As a Burundian informant put it, Congolese people are seen as more likely to 'engage with HF staff' but also to 'muddle through' in the absence of the State. This idea is commonly referred to in DRC as 'article 15', the imaginary Article 15 of the Congolese Constitution stating 'débrouillez-vous' ('fend for yourself' or 'get on with it' [48].

Heterogenous effects are a possible way to formally explore these hypotheses using our quantitative data. Our dataset presents within as well as between country variation at the level of each HF/HFC; it is, therefore, possible to compare the explanatory value of the Kivu interaction effect with an alternative interaction effect (see the evaluation framework in S1 Text. Formally, we use model 4 as presented in S1 Text, replacing K with another source of heterogeneity). We focus on the effects on the second dimension, social accountability at the HF-level, as it is the first objective of the intervention and a dimension that is substantially different between the two cases.

The first hypothesis, the composition of the HF, can be tested using different indicators: the level of education, profession, and gender ratio of HFC members. Only the ratio of committee members with secondary education proves a (mildly) conclusive hypothesis (see S7 Table): the interaction is signification but weaker and a much less good fit (adjusted r-squared) than in the case of the model with the Kivu binary variable.

The second hypothesis, the management and incentive structure, is present in Fig 4 (details are found in tested in S8 Table). We create a binary variable indicating whether the HF is faith-based, but also distinguish between Catholic and Protestant faith-based HFs. The reason is that Catholic HFs are known to have a more rigid and centralised form of governance that makes them function similarly to fully public health centres [49]. Catholic HF would typically depend on a religious congregation or the diocese headquartered far away from the HF. Anglican and, to an even larger extent, Evangelical, Pentecostal, Baptist, and Adventist health centres are typically dependent on a more local church.

The interaction is strongly significant in the case of the Protestant HF. It is important to note, however, that faith-based HF often started in a worse-off situation (see S8 Table). Our main hypothesis is that, in a multiple principal-agents scenario where the HF (staff) has the health district, the HFC, and the church as principals, the intervention reinforces the place of the HFC. This is because the health district gives faith-based HF more autonomy in management *and* because the church, which owns the HF, is open to direct community participation. Another possible explanation is that there is a natural community of church-goers in the vicinity who may take an interest in the management of their local HF. This hypothesis is in line with public health work in Cambodia [50], which shows a greater mobilisation around pagoda-related HF than around lay HF. It is not necessarily incompatible with the management hypothesis. One needs to be careful, however, as the presence of a faith-based HF may not mean that there is a church of the same faith in the immediate vicinity. The medical staff of faith-based HFs are not necessarily directly related to the clergy of the church that owns the HF.

The third hypothesis is the idea that a stronger tradition of participating in decision-making about public goods in South Kivu makes the intervention a better fit there. In Burundi research depicts a centralised society with a rigid structure of control that goes down to the local level, with people fearing to challenge from administrative, security, or political authorities [51]. Nkurunziza shows how service provision has been instrumented for a long time and is a function of the unrepresentative elite's preferences rather than the product of social struggle [52]. Ndikumana (2007) underlines that there is no history of tribal authority in Burundi and that

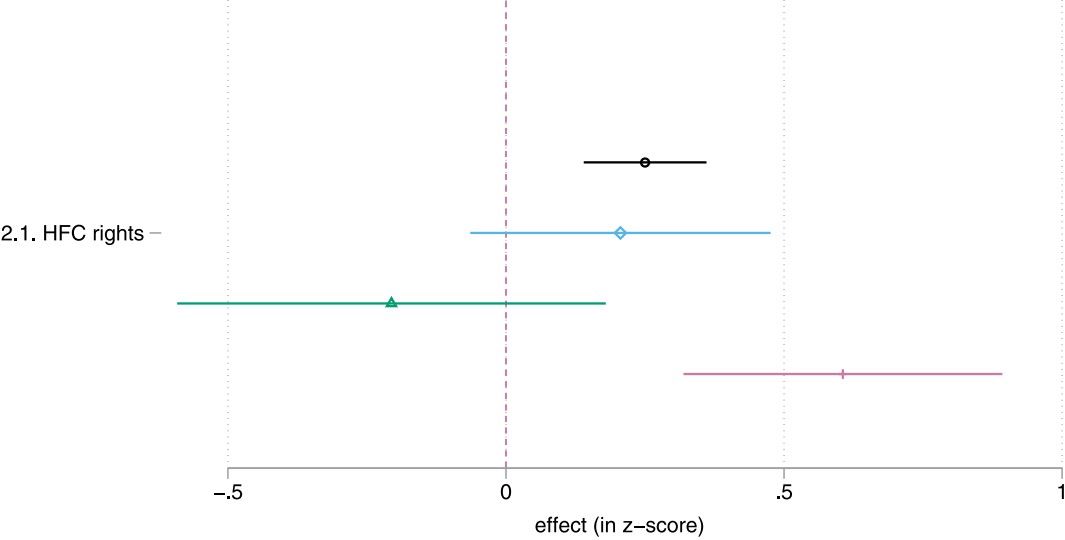

2.1. HFC rights

effect (in z-score)

- ○ accross HFs
- ◇ in faith-based HFs (vs faith-based HFs without intervention)
- △ in Catholic HFs (vs Catholic HFs without intervention)
- + in Protestant HFs (vs Protestant HFs without intervention)

**Fig 4. Intent-to-Treat—HF management (diff-in-diff, without controls), by country.** Note: lines represent a 95% confidence interval for the estimate. Estimates whose lines cross the red vertical line are not statistically different from zero. Detailed results are found in S8 Table (ANCOVA model). | Please see the methods section and S1 Table for details on the sampling (estimates use the full sample, n = 329 HFC/HF) and data sources.

"the power structures guarantee that political dissent is suffocated and does not trickle through the layers of institutional fences to disturb the central authority. At the same time, the population remains alienated and silenced (*muselée*)" [53] (also see [54]. There is, however, important regional variation [55]).

Across the border, the Mashi and Muhavu cultures have been described as 'in the mist', at a distance from central power [56]. Research has highlighted a vocal civil society and a more direct attitude of confrontation with the authority and resource mobilisation within the community [57]. The central power and party politics that characterise Burundi are less present in South Kivu, which was historically the borderland of centralised and occasionally hostile political entities (such as the kingdom of Rwanda for the health zones of this study). In a region that became largely abandoned by the central government at the end of the twentieth century, health and school services were often left to the hands of local communities [58].

A set of questions from the endline population survey helps to understand further how people relate to authority, State, and service delivery (Table 3). There is no difference between the intervention and control groups. Burundians are, on average, expecting more from the State and avoiding direct confrontation with HF staff, which confirms the anthropological and historical literature. These perceptions are partly informed by past and practical experiences but also echo a more general attitude towards public services and even a different understanding of interpersonal relationships. As one interviewee put it, often 'in Burundi, we want to preserve good relationships. You never know when you could need somebody, so you do not want to hurt or shock, and you hold back to preserve relationships'.

**Table 3. Relationship to state and service providers (descriptive statistics).**

|  | (1) | (2) | (3) | N |
|---|---|---|---|---|
|  | Burundi | Sud Kivu | difference (p-value) |  |
| Trusts the state to provide basic social services such as healthcare or education (0–4) | 3.153 (0.033) | 1.695 (0.078) | 0.000 | 9525 |
| What would you do if you have a problem at your local HF? (multiple answers) |  |  |  |  |
| Talk to administration | 0.036 (0.005) | 0.036 (0.006) | 0.933 | 8568 |
| Tak to local leaders | 0.022 (0.004) | 0.092 (0.011) | 0.000 | 8568 |
| Talk to religious leader | 0.003 (0.001) | 0.048 (0.008) | 0.000 | 8568 |
| Talk to HFC | 0.105 (0.009) | 0.315 (0.023) | 0.000 | 8568 |
| Face HF staff | 0.047 (0.004) | 0.264 (0.018) | 0.000 | 8568 |
| Complain to neighbours, spread the word | 0.319 (0.016) | 0.064 (0.013) | 0.000 | 8568 |
| Nothing, I would be stuck | 0.473 (0.017) | 0.231 (0.022) | 0.000 | 8568 |

Integrating this element into our evaluation framework is challenging as the data is not meant to be representative at the HF-level. As such, the district (between 8 and 23 HFs) may be too high a level of aggregation for exploring local relationship to authority. Nevertheless, we can create a proxy of the level people feel they can confront service providers by generating HF- and district-level indicators reflecting the weighted proportion of survey respondents who indicated that they would face the HF staff if they are unhappy with the services (see S8 Table). Both indicators have clear shortcomings, but it is worth noting that their interaction with the treatment is positive in both cases (S9 Table).

None of the contextual factors ─HFC education, HF management, or relationship to authorities─ single-handedly explain the Kivu interaction effect, but it is worth noting that adding only two new sources of heterogeneity to the model lowers the value and significance level of the Kivu interaction effect substantially (S10 Table). As highlighted earlier, there are clear limitations with this approach. Nevertheless, it allows us to cautiously characterise what in the context matters: in this case, mostly the type of management of the HFC (faith-based or not) as well as local, long-running relationships with authorities and service providers. Breaking down the context in the way we have attempted to do suggests directions for understanding the type of external validity claim that can be made.

## 6. Conclusion

In South Kivu, but not in Burundi, a simple training intervention strengthened the position of Health Facility Committees vis-à-vis their health facility nurses and improved the management of the facilities. We argued that two elements are central to understanding such discrepancy: (1) the type of management of social services and their anchoring in the local tissue, and (2) local patterns and histories of relationship to authorities and service providers. While the former aspect speaks to the health management and health economics literature, the latter echoes work in institutional economics [59]. Both points are ultimately about the room available to reshuffle the HFC's property rights (their ability to make decisions at the HF), and confirm the idea that 'changing the [current] equilibrium [. . .]—is at the heart of effective participatory development' [2]. Importantly, our paper has also highlighted the limitations of community governance reforms which do not, over the 12–18 months of the study, substantially affect the quality and use of services. Of course, it does not mean that they may not contribute to creating dynamics that have longer-run consequences.

The findings bring nuance to the debates on community governance in healthcare and basic social services more generally. They confirm evidence on the limits [29] and the

heterogeneous nature of bottom-up accountability mechanisms [6]. Our findings are slightly more optimistic than Humphreys et al.'s study of participatory development committees set up by international aid in Eastern DRC [60], whether this is due to looking at different sectors or different institutions–the HFCs are old institutions that pre-date committees set up by international aid as part of community-driven reconstruction initiatives–will need to be explored further. In line with them, though, we find some disconnect between committees and the population they represent. In other terms, the power that is gained by the committees in South Kivu seems to be primarily for themselves. Whether this constitutes good enough, and meaningful, bottom-up accountability in a complicated context such as South Kivu is an open question, and certainly one Ministries of Health and aid organisations should ask themselves.

## Supporting information

**S1 Table. Indicators for impact evaluation.**
(DOCX)

**S2 Table. Situation at baseline.**
(DOCX)

**S3 Table. Intent-to-Treat—Main effects.**
(DOCX)

**S4 Table. Intent-to-Treat—Main effects, by country.**
(DOCX)

**S5 Table. Intent-to-Treat—Effects for the social accountability indicators at HF-level (HF rights).**
(DOCX)

**S6 Table. Intent-to-Treat—Indicators of HF management (robustness checks).**
(DOCX)

**S7 Table. Heterogeneous effects: HFC composition (ANCOVA).**
(DOCX)

**S8 Table. Heterogeneous effects: Management structure (ANCOVA).**
(DOCX)

**S9 Table. Heterogeneous effects: Relationship to HF (ANCOVA).**
(DOCX)

**S10 Table. Heterogeneous effects—Multiple sources (ANCOVA).**
(DOCX)

**S1 Text. Evaluation framework.**
(DOCX)

## Acknowledgments

We are very grateful to Pierre Bakevya, Bernard Nyagashahu, Freddy Irambona, Isidore Nkunzimana, and Paterne Kalegemire who led the implantation of the interventions, as well as to Placide Nibogora, Egide Nduwimana, Jean-Paul Zibika and their teams of brilliant enumerators. Stefan Dercon, Kate Orkin, Olivier Sterck, Winnie Yip, Nicolas Van de Sijpe, Doug Gollin, Biju Rao, Macartan Humphrey, and Joëlle Schwarz provided excellent suggestions on this project and earlier iterations of the manuscript. The research design, analysis plan, and

findings were presented on different occasions; we thank participants to workshops and conferences in Oxford (ODID, CSAE, and Nuffield), Cambridge (African Studies), Sussex (ASA conference), Berlin (BSSSS), Montreal (Public Health), Bukavu (Ministry of Health), and Bujumbura (Ministry of Health). Thank you to Clayton Boeyink for proof-reading the manuscript.

## Author Contributions

**Conceptualization:** Jean-Benoit Falisse, Hugues Nkengurutse, Léonard Ntakarutimana.

**Data curation:** Jean-Benoit Falisse, Léonard Ntakarutimana.

**Formal analysis:** Jean-Benoit Falisse, Hugues Nkengurutse, Léonard Ntakarutimana.

**Funding acquisition:** Jean-Benoit Falisse.

**Investigation:** Jean-Benoit Falisse, Hugues Nkengurutse, Léonard Ntakarutimana.

**Methodology:** Jean-Benoit Falisse, Hugues Nkengurutse, Léonard Ntakarutimana.

**Project administration:** Jean-Benoit Falisse, Léonard Ntakarutimana.

**Resources:** Jean-Benoit Falisse.

**Software:** Jean-Benoit Falisse.

**Supervision:** Jean-Benoit Falisse, Hugues Nkengurutse, Léonard Ntakarutimana.

**Validation:** Jean-Benoit Falisse.

**Visualization:** Jean-Benoit Falisse.

**Writing – original draft:** Jean-Benoit Falisse.

**Writing – review & editing:** Jean-Benoit Falisse, Hugues Nkengurutse, Léonard Ntakarutimana.

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
