## [Decision Letter · Decision Letter 0]

24 May 2023

PGPH-D-23-00145

Strengthening the Community Governance of Health-care Services in ‘Fragile’ Settings: Evidence from Burundi and South Kivu, DR Congo

Dear Dr. Falisse,

Thank you for submitting your manuscript to PLOS Global Public Health. After careful consideration, we feel that it has merit but does not fully meet PLOS Global Public Health’s publication criteria as it currently stands. Therefore, we invite you to submit a revised version of the manuscript that addresses the points raised during the review process.

We look forward to receiving your revised manuscript.

Kind regards,

Hassan Haghparast Bidgoli

Academic Editor

Journal Requirements:

1. Please send a completed 'Competing Interests' statement, including any COIs declared by your co-authors. If you have no competing interests to declare, please state "The authors have declared that no competing interests exist". Otherwise please declare all competing interests beginning with twhe statement "I have read the journal's policy and the authors of this manuscript have the following competing interests:"

2. We have noticed that you have uploaded Supporting Information files, but you have not included a list of legends. Please add a full list of legends for your Supporting Information files after the references list. 

Additional Editor Comments (if provided):

Please address the issues raised by the reviewers.

- I agree with the reviewer 2 in regards to clarifying the research design. Please add more information on the design

- Please separate results and discussion sections

- Your research was conducted in Burundi and DR Congo but ethical approval only obtained from Burundi. Please explain why you donor have ethics approval from DR Congo.

Reviewers' comments:

Reviewer's Responses to Questions

**Comments to the Author**

1. Does this manuscript meet PLOS Global Public Health’s publication criteria? Is the manuscript technically sound, and do the data support the conclusions? The manuscript must describe methodologically and ethically rigorous research with conclusions that are appropriately drawn based on the data presented.

Reviewer #1: Yes

Reviewer #2: Partly

2. Has the statistical analysis been performed appropriately and rigorously?

Reviewer #1: I don't know

Reviewer #2: I don't know

3. Have the authors made all data underlying the findings in their manuscript fully available (please refer to the Data Availability Statement at the start of the manuscript PDF file)?

Reviewer #1: Yes

Reviewer #2: Yes

4. Is the manuscript presented in an intelligible fashion and written in standard English?

Reviewer #1: Yes

Reviewer #2: Yes

5. Review Comments to the Author

Reviewer #1: This is not only a very relevant paper, it is also a very nicely written one (with however here and there some small typos - a thorough "re-reading" may suffice to address this issue).

The sections on introduction and rationale and literature are excellent.

Statistics are not at the center of my area of expertise and I can therefore not express a fully confident opinion on the quality of their statistical methods and analysis. I hope that the other reviewer(s) can address this.

The results section (that in fact includes also the discussion) is again very interesting. The interpretation of the findings - especially when it comes to analyze the importance of context in explaining the difference in results between South-Kivu Province in DRC and Burundi) - make "a lot of sense" to me. They are indeed "congruent" with my own field experience as public health professional familiar with the two contexts described.

This paper absolutely deserves to be published. I learned from it and congratulate the authors with this excellent piece of work.

Reviewer #2: This is an interesting study contributing to the limited evidence base on community governance and its contribution to improving health systems’ functioning in fragile settings. While it presents relevant findings, the structure and narrative of the paper needs to be improved to allow to the reader a better understanding of the study context, methods and results/discussions. More specific recommendations can be found below:

- In this version, the reader can find results in the introduction as well as discussion elements in the results. Please make sure to move all results to the corresponding section (examples include lines 61 – 65 “In Burundi, only the organization of the HFC … quality of services” and table 1) for ease of reading and understanding among readers.

- Some statements related to the context characteristics (e.g. first paragraph of section 3) are presented with any references. Please add relevant citations.

- The presentation of methods requires some clarifications/modifications. The main concern is the lack of clarity about the research design of the quantitative component. Authors refers to it as an RCT (randomized clinical? trial) but they do not present all the details: how was the randomization process conducted/secured? What is the ratio given that 168 vs 83 facilities were selected (vs not) in Burundi? Authors are invited to explain this more or to identify this component as non-randomized trial if applicable.

- The evaluation framework can be an appendix unless presented in simplified way to readers with no strong statistics background. The actual data collection methods especially surveys need to be clarified (with tools or variables can be added in the appendices) as the paragraph under table 2 is not clear. For instance, which survey informed table 3 results? What is the sample size of this (or these) surveys? How was data analysed? Also, does table 3 shows no statistical difference between the two groups for all variables as the footnotes show an explanation of level of significance for (*), (**?) and (***) but without using these signs in the table. Could you please mention it explicitly if none of the comparison had a p<0.1)

- On the main outcomes, there is a confusion whether the study included 2 or 3 outcomes under accountability, as table 2 mentions 3 but the subsequent tables mention HFC org. and HFC reach. Overall, the tables are difficult to read.

- The results section includes discussion of many findings with reference to other studies in the literature. Authors may want to consider writing a discussion section where these points are noted or (in case of a reason for not to) to call their results section as “results and discussion”.

6. PLOS authors have the option to publish the peer review history of their article (what does this mean?). If published, this will include your full peer review and any attached files.

**Do you want your identity to be public for this peer review?** For information about this choice, including consent withdrawal, please see our Privacy Policy.

Reviewer #1: **Yes: **Bart Criel

Reviewer #2: No

---

## [Decision Letter · Decision Letter 1]

17 Jul 2023

Strengthening the Community Governance of Health-care Services in ‘Fragile’ Settings: Evidence from Burundi and South Kivu, DR Congo

PGPH-D-23-00145R1

Dear Mr Falisse,

We are pleased to inform you that your manuscript 'Strengthening the Community Governance of Health-care Services in ‘Fragile’ Settings: Evidence from Burundi and South Kivu, DR Congo' has been provisionally accepted for publication in PLOS Global Public Health.

Best regards,

Hassan Haghparast Bidgoli

Academic Editor

Thanks for addressing all the comments raised by the reviewers and the academic editor. There are no further comments from the reviewers and the editor.

Reviewer Comments (if any, and for reference):

Reviewer's Responses to Questions

**Comments to the Author**

1. If the authors have adequately addressed your comments raised in a previous round of review and you feel that this manuscript is now acceptable for publication, you may indicate that here to bypass the “Comments to the Author” section, enter your conflict of interest statement in the “Confidential to Editor” section, and submit your "Accept" recommendation.

Reviewer #1: All comments have been addressed

Reviewer #2: All comments have been addressed

2. Does this manuscript meet PLOS Global Public Health’s publication criteria? Is the manuscript technically sound, and do the data support the conclusions? The manuscript must describe methodologically and ethically rigorous research with conclusions that are appropriately drawn based on the data presented.

Reviewer #1: Yes

Reviewer #2: Yes

3. Has the statistical analysis been performed appropriately and rigorously?

Reviewer #1: I don't know

Reviewer #2: Yes

4. Have the authors made all data underlying the findings in their manuscript fully available (please refer to the Data Availability Statement at the start of the manuscript PDF file)?

Reviewer #1: Yes

Reviewer #2: Yes

5. Is the manuscript presented in an intelligible fashion and written in standard English?

Reviewer #1: Yes

Reviewer #2: Yes

6. Review Comments to the Author

Reviewer #1: The authors have adequately addressed the comments I formulated in the initial review.

Reviewer #2: Thank you for addressing my comments. The current version reads well and presents the study in a very good shape.

7. PLOS authors have the option to publish the peer review history of their article (what does this mean?). If published, this will include your full peer review and any attached files.

**Do you want your identity to be public for this peer review?** For information about this choice, including consent withdrawal, please see our Privacy Policy.

Reviewer #1: **Yes: **Bart Criel

Reviewer #2: No
